# Extracts and Residues of Common Ragweed (*Ambrosia artemisiifolia* L.) Cause Alterations in Root and Shoot Growth of Crops

**DOI:** 10.3390/plants12091768

**Published:** 2023-04-26

**Authors:** Rea Maria Hall, Helmut Wagentristl, Katharina Renner-Martin, Bernhard Urban, Nora Durec, Hans-Peter Kaul

**Affiliations:** 1Institute of Agronomy, University of Natural Resources and Life Science, Vienna, 3430 Tulln an der Donau, Austria; noradurec@gmx.at (N.D.); hans-peter.kaul@boku.ac.at (H.-P.K.); 2Institute of Botany, University of Natural Resources and Life Science, Vienna, 1180 Vienna, Austria; bernhard.urban@boku.ac.at; 3Experimental Farm, University of Natural Resources and Life Sciences, Vienna, 2301 Groß-Enzersdorf, Austria; helmut.wagentristl@boku.ac.at; 4Institute of Mathematics, University of Natural Resources and Life Science, Vienna, 1180 Vienna, Austria; katharina.renner-martin@boku.ac.at

**Keywords:** common ragweed, aqueous root extracts, aqueous shoot extract, essential oil, residues, rhizobial nodules

## Abstract

Following the novel weapon hypothesis, the invasiveness of non-native species, such as common ragweed (*Ambrosia artemisiifolia* L.) can result from a loss of natural competitors due to the production of chemical compounds, which negatively affect native communities. Particularly the genus Ambrosia produces several types of organic compounds, which have the potential to inhibit germination and growth of other plants. Subsequent to an assessment of the chemical content of three different ragweed extracts (aqueous shoot and root extracts, as well as essential oil), two different trials on the effects of different concentrations of these extracts, as well as ragweed residues, were conducted on two different mediums (Petri dish vs. soil). In addition, we investigated the impact on the infection potential of *Bradyrhizobium japonicum* on soybean roots in three different soil types (arable soil, potting soil, and sand). The results showed that the exposure to common ragweed extracts and residues induced changes in the biomass and root production of crops and ragweed itself. Even though crops and ragweed differed in their response behavior, the strongest negative impact on all crops and ragweed was observed with ragweed residues, leading to reductions in biomass and root growth of up to 90%. Furthermore, we found a decrease in the number of rhizobial nodules of up to 48% when soybean was exposed to ragweed root extract.

## 1. Introduction

Invasion by non-indigenous species is one of the major drivers of restructuring and malfunctioning of ecosystems and can pose threats to human health, economy, and agriculture [1,2,3]. An important role in understanding the invasiveness of introduced exotic species is often attributed to their release from their native environment. As proposed by the enemy release hypothesis [4], introduced plants are liberated from their specialist herbivores and pathogenes, hitherto gaining a competitive advantage over natives. Resources, which are usually invested in costly traits that confer resistance to natural enemies, can be re-allocated to the development of other traits, which could constitute an advantage over native plants in the invaded range, such as size or fecundity [5,6,7]. Besides morphological traits, biochemical constituents of invasive species may also contribute to their success. The novel weapons hypothesis [8] proposes that some invaders owe their success to the production of biochemicals that may have the potential to exert stronger effects on native species that lack a coevolutionary-based tolerance than on coevolved competitors in the native range that had adapted over time. This chemical interference between plants was first described as allelopathy by [9]. In this context, the annual herbaceous common ragweed (*Ambrosia artemisiifolia* L., subsequently named ragweed), which is native to North America, but which has migrated to farm fields and early successional sites in Europe since the mid-20th century, is of particular interest [10,11,12]. In several countries, e.g., Austria, Germany, Hungary, France, Italy, Ukraine, Russia, Serbia, Croatia, Slovenia, and Switzerland, common ragweed is nowadays one of the most important agricultural weeds, causing severe yield losses in soybean, maize, or sunflower [13,14,15]. Furthermore, it is also a main threat to human health due to abundant allergenic pollen release, which was identified to be the main cause of hay fever and allergic rhinitis in late summer in Central Europe [16,17,18]. Management options to effectively contain the spread of the plant are limited due to its plasticity, its highly adaptive behavior in terms of growing conditions, and its tolerance against herbicides and mechanical damage [11,19,20,21]. Different studies on the allelopathic effects of common ragweed revealed that this plant species produces and releases several types of secondary metabolites, including the sesquiterpene ambrosic acid, phenols, thiarubrines, and thiopenes [22,23,24]. Most of these compounds have a broad spectrum of biological activities, and therefore have the potential to inhibit germination and growth of other plants and microorganisms [24,25,26]. For example, Ref. [15] showed that the presence of ragweed decreased the nodulation performance of nitrogen fixing rhizobial bacteria on soybean and runner bean plants substantially, causing a severe decrease in biomass production and yield.

Recently, most of these studies were executed using only residues of the plant, only one or two extracts, and only one medium. Ref. [26] showed that ragweed residues in soil significantly affected seed emergence and seedling growth of different indicator crops, such as wheat and tomato. Laboratory trials showed that the presence of an aqueous extract of aboveground and belowground dry matter of the plant reduced germination rate of maize [27]. Methanol and hexane extracts of aboveground plant parts of ragweed led to lower germination rates in cress, radish, and red clover [28]. However, information on the chemical compounds and how their effect is promoted or mitigated under different growing conditions (Petri dishes vs. soil) is very low. Thereby, it has to be taken into account that different extracts have different chemical compounds, depending on the polarity of the solvent. In addition, they can be subject to natural fluctuations, depending on various biotic and abiotic factors during the development of the plant [29,30,31].

In order to obtain a better insight in the chemical interference between ragweed and crops, which are common in Central Europe’s field rotations, the aim of the present study was (1) to assess the chemical profile of the aboveground and belowground biomass of common ragweed in three different extracts (aqueous extract of ragweed shoot and roots, and essential oil). Furthermore, we investigated (2) the effects of different concentrations of these extracts, as well as of different concentrations of ragweed residues on germination and seedling development of three different crops (soybean, wheat, and maize) and of ragweed itself in two different growing media (Petri dishes vs. soil). In an additional experiment, (3) the impact of ragweed roots and aqueous extracts of ragweed roots on the nodulation potential of *Bradyrhizobium japonicum* on soybean was observed in three different soil types (arable soil, potting soil, and sand) to detect possible inhibition effects.

## 2. Results

### 2.1. Chemical Compounds of Common Ragweed

In total, we found 57 different chemical compounds in the essential oil, mainly terpenes and sesquiterpenes. With 48.9%, germacrene D was the most abundant compound, followed by germacrene B (7.8%) and β-caryophyllene (3.8%). The 20 most abundant chemical constituents are summarized in Appendix A. The analytical characterisation of the aqueous shoot and root extracts led to the identification of multiple sesquiterpenoid lactones, i.a., artemisinin, psilostachyin, and isabelin. The results of this analysis can be found in the Appendix A.

### 2.2. Laboratory Experiment: Ragweed Had Significant Impact on Root Growth

#### 2.2.1. Germination

The germination rate of the crops was nearly unaffected by the presence and the concentration of ragweed extracts and residues (Appendix A). In contrast, ragweed seeds exposed to root extracts (RT1-100) of conspecifics showed an approx. 32% lower germination rate than the untreated control seeds (80%), independent of the concentration. Additionally, the presence of 10% residues (Resi10) decreased the germination rate to 55%.

#### 2.2.2. Root Length

Seven days after germination, we found significant alterations in root growth of crops and ragweed seedlings when exposed to the various ragweed extracts and/or ragweed residues (Appendix A). Compared to the untreated control, the root length of soybean seedlings was decreased by approx. 54% on average when exposed to shoot extracts (ST1/10/100), independent of the concentration (Figure 1a). Roots exposed to 1% root extracts (RT1) were 25% shorter, while increased concentrations (RT10 and RT100) led to a 59% and 50% reduction in root growth. The exposure to essential oil caused a decrease of 18% (EO0.5) and 21% (EO1) in soybean root length, whereas the exposure to low concentrations of residues (Resi1), even slightly, promoted root growth by 6%. With wheat, a low concentration of shoot extracts (ST1) did not have any effect on the root length, but, when seedlings were exposed to ST10 and ST100, root growth decreased significantly by 25% and 64%, respectively (Figure 1b). The same trend was observed with wheat seedling roots exposed to root extracts of ragweed. Particularly, the exposure to RT100 reduced root growth by 53%. However, the most significant impact was the exposure of wheat seedlings to 1% essential oil of ragweed (EO1), which decreased the root length by 78%. As with soybean, we also noticed a slight increase of 7% in root growth when wheat seedlings were exposed to Resi1, but this trend reversed when wheat seedlings were exposed to higher concentrations of residues.

Maize seedling roots were not affected by root extracts and residues, but they developed 33% and 41% shorter roots when exposed to ST10 and ST100, respectively. A contrasting effect was observed with ragweed essential oil. Whereas, maize roots exposed to EO0.5 showed a 62% decrease in length, the exposure to EO1 reduced root growth only by 28% (Figure 1c). In contrast to the crops, ragweed seedlings were not affected by extracts of conspecifics, even though there was a trend wherein the exposure to low and medium concentrations of shoot extracts (ST1 and ST10) promoted root growth by 27% and 62%, respectively. Nevertheless, a significant impact was only measured when ragweed seedling roots were exposed to residues of ragweed. The presence of low concentrations of residues (Resi1) was enough to decrease root length by 50%. When exposed to Resi10, ragweed roots were 67% shorter than the untreated control plants (Figure 1d).

### 2.3. Greenhouse Experiment 1: Soil Conditions Enhanced the Effects of Ragweed Extracts and Residues

Soil conditions enhanced the effects of ragweed extracts and residues on biomass and root production: aboveground dry matter (AGDM) and root dry matter (RDM) of crops and ragweed were measured 10, 17, and 24 days after seeding (DAS). All results are summarized in Appendix A. Figure 2a–d and Figure 3a–d summarize the results at 24 DAS, which depicted the influence of the extracts most clearly. As in the laboratory experiment, we detected no influence of the ragweed extracts on the germination rate, but, throughout the trial, we found significant impact of the ragweed extracts and residues on biomass and root production of the crops and ragweed itself.

#### 2.3.1. Ragweed Shoot Extract

During the first 17 days, the exposure to shoot extracts (ST) had a positive effect on the biomass production of soybean. 10 DAS plants exposed to ST1 showed a 41% higher AGDM than the untreated control plants. The biomass of soybean plants exposed to ST10 and ST100 was increased by 22% on average. However, this trend reversed, and after 24 days we observed a 47% reduced biomass production at medium extract concentrations (ST10) and a 50% decrease in AGDM at high ST100. Root growth was not affected during the first ten days, but already, after 17 days of exposure, we observed a significant decrease in root mass of 36% at ST100. This trend was enhanced for 24 DAS plants. At the last sampling, roots exposed to ST10 and ST100 had 64% less root mass than the unexposed control. With wheat and maize, we found a clearly promoting effect of the ragweed shoot extract (ST) on the AGDM production of both crops until 17 DAS. This effect dissolved with wheat until 24 DAS. Opposingly, we found a clear reduction effect on the root biomass. Wheat plants treated with ST10 produced 25% less root mass, and plants exposed to ST100 lost 37% of root mass compared to the untreated control plants. In contrast, after 24 days of exposure, maize plants showed a significant reduction in AGDM of 36% at ST100 but, unlike wheat, maize root mass, increased significantly by 49%, especially at medium extract concentrations (ST10). As opposed to the crops, the biomass production of ragweed was significantly enhanced when exposed to shoot extracts of conspecifics. Particularly, medium and lower concentrations of the extract (ST10 and ST100) led to threefold more biomass than the untreated control plants. The same positive effect was observed with the root mass. Ragweed plants exposed to ST10 produced substantially more root mass than the untreated control plants.

#### 2.3.2. Ragweed Shoot Extract

With the ragweed root extract (RT), we observed similar effects as with the shoot extract. During the first 10 days, AGDM of soybean was significantly enhanced by approx. 40% when exposed to low (RT1) and medium concentrations (RT10). After 24 days, this trend inverted, and soybean plants produced approx. 60% less biomass when exposed to RT10 and RT100 than the control plants. The same effect was observed with the soybean root mass, which was not affected during the first 17 days of the experiment. After 24 days, the exposure to RT10 and RT10 caused a severe decline in root mass of 52% and 79%, respectively. Throughout all sampling dates, wheat AGDM was not affected by ragweed root extracts, even though the root mass of wheat was substantially reduced by 31% after 24 days. Opposingly, maize showed a significantly reduced biomass production (−36% and −46%, respectively), particularly when exposed to low concentrations (RT1) and high concentrations (RT100), but it did not show any changes in root mass production. Other than the shoot extracts, the exposure to root extracts also affected ragweed biomass production negatively. Until 17 DAS there were no significant effects to notice. However, after 24 days, the AGDM of ragweed was reduced by 27% when exposed to RT10 and even 65% when exposed to RT100. This development was also depicted by the ragweed root mass, particularly at high root extract concentration. Plants exposed to RT100 had 79% less root mass than the untreated control plants.

#### 2.3.3. Ragweed Essential Oil

Unlike the aqueous extracts, we found significant impact on all crops and ragweed itself when exposed to ragweed essential oil. Even though there were promotional effects of low oil concentrations (EO0.5) on the biomass production of soybean and maize until 17 DAS, after 24 days, all crops showed significantly lower AGDM. Lower concentrations of ragweed essential oil (EO0.5) were enough to reduce soybean AGDM by 46%, and, when exposed to EO1 soybean plants produced 64% less biomass. A similar effect was observed with soybean roots. After 24 days of exposure, soybean root mass was 45% lower at EO0.5 and 65% lower at EO1. A similar effect was observed with wheat, with which the exposure to EO1 caused a substantial reduction in AGDM by 75% and led to a decrease in root mass by even 76%. Whereas, the exposure to EO0.5 did not have any effect on the AGDM production of wheat, this concentration was enough to reduce root production by 36%. Irrespective of the concentration, the presence of ragweed essential oil did not have any effect on the root mass production of maize, but, already, low concentrations (EO0.5) caused a loss in AGDM of 39%. However, this biomass loss was less significant at high oil concentrations (EO1) and accounted for 24%. The same phenomenon was observed with ragweed. The exposure to essential oil had no effect on the root production, but the treatment with EO0.5 caused a reduction in AGDM by 69%, whereas EO1 caused only a loss of 33%.

#### 2.3.4. Ragweed Residues

In contrast to the laboratory experiment, the most severe impact on biomass and root production of crops and ragweed was observed with ragweed residues. Soybean did not show any changes in biomass production until 17 DAS, but, afterwards, AGDM production was severely reduced by 55% averaged over all concentrations. Even more severe was the impact on the root mass. Already at low residue concentration, (Resi1) caused a root mass decrease of 79%. Additionally, wheat was not affected during the first sampling dates, but then it showed a decrease in AGDM production by 31% when exposed to high residue concentration (Resi10). While Resi5 did not have any effect on the AGDM production, this medium residue concentration caused a significant reduction in root mass by 53%. At high concentrations (Resi10), wheat produced 59% less root mass. Maize AGDM was decreased from the outset when exposed to Resi10. Already, after 10 days, maize bio-mass was 54% lower than that of the unexposed control plants. After 24 days, plants exposed to Resi10 produced even 87% less biomass. However, lower concentrations of ragweed residues also led to significant decreases in AGDM by 36% (Resi1) and 61% (Resi5). The impact on the root mass was slightly attenuated, but it was also significant. After 24 days, the exposure to low concentrations (Resi1) had a small promotional effect of 8%, but the treatment with Resi5 and Resi10 caused a decline in root mass by 28% and 34%, respectively.

The highest reduction in biomass production was observed with ragweed itself. Lower residue concentrations (Resi1) did not have any noticeable effect. However, after 24 days, plants exposed to Resi5 produced 90% less biomass, and the AGDM of plants exposed to Resi10 lost 92% of biomass compared to the untreated control. The impact on the root mass was even more severe. Ragweed plants exposed to Resi1 already lost 83% of root mass. The treatment with Resi10 caused a root mass decrease of 98%. Due to this severe impact of ragweed residues, which was observed evenly in all crops and ragweed, we fitted dose–response curves (Appendix A) based on the root length, which correlated significantly with the root biomass (r = 0.89, *p* < 0.001). In addition, no data transformation was necessary with this parameter. The lowest dosages of ragweed residues required to reduce root length growth by 20% (ED20) and 50% (ED50) were calculated for soybean and ragweed itself. A residue concentration of 0.5% and 0.6%, respectively, was enough to reduce soybean and ragweed root length by 20% (ED20). At a dosage of 1.6% and 1.4% respectively, root length of soybean and ragweed decreased by 50% (ED50). With an ED20 of 4.8% and an ED50 of 7.2% residue concentration, maize was less sensitive to the ragweed residues. Additionally, wheat was a little bit more persistent than soybean and ragweed, accounting for an ED20 of 0.8% and an ED50 of 2.4%.

### 2.4. Greenhouse Experiment 2: Roots and Root Extracts Reduce Nodulation of Soybean, Irrespective of the Soil Type

Figure 4 depicts the results of the second greenhouse trial, indicating that not only the root extracts, but also root residues of ragweed, severely affect the nodulation potential of *Bradyrhizobium japonicum*, irrespective of the soil type. The most severe decrease in the number of nodules was observed with arable soil contaminated with ragweed root extract (AqR). Only 18 ± 10 nodules per plant were found on average, compared to the untreated control plants which were infested by 44 ± 9 nodules on average. Root residues (root) led to reduction of 44% (25 ± 8 nodules) when incorporated in arable soil. The highest number of nodules of 50 ± 9 was found on untreated soybean plants growing in potting soil. The contamination of this potting soil with ragweed roots decreased the number of nodules to 26 ± 8 (−48%). Root extracts of ragweed caused a 30% reduction (30 ± 10 nodules). The most inert soil type was sand, where we observed no difference between the control plants (28 ± 5 nodules) and plants grown with root residues (20 ± 11 nodules). However, we found a significant decrease of 43% less nodules (15 ± 7) when exposed to root extracts of ragweed.

## 3. Discussion

The results of the present study revealed that the exposure to common ragweed extracts and residues induced changes in the biomass and root production of crops and ragweed itself. Furthermore, we found severe negative impact on the nodulation potential of the nitrogen-fixing *Bradyrhizobium japonicum*. It was shown that the effects of these extracts and residues vary in dependency of the medium (Petri dish vs. soil) used in the experiments, leading to a mitigation or an enhancement of these impacts. However, we found clear differences in the response behavior between crops (wheat, soybean, and maize) and also ragweed itself, pointing out the importance of plant–plant interaction mediated by chemical compounds [32,33].

In none of our experiments the germination of crops was affected by the presence of ragweed extracts or residues, which is controversial with regard to numerous studies that deal with the inhibitory impact of ragweed on the germination performance of crops, i.e., [22,26,27]. In contrast, ragweed germination was severely reduced by high amounts of residues and by the presence of root extracts of conspecifics. This autotoxicity of ragweed mediated by root exudates was first shown by [34]. It is seen as an important means of population density regulation, as postponing germination may avoid establishing in an area/environment where intraspecific competition would prevent their survival. Thus, it strengthens resource competition and enables invasive success [6]. Nevertheless, when ragweed root extracts were incorporated in soil (greenhouse trial), this effect on conspecifics germination was not observable.

### 3.1. Effects of the Ragweed Extracts and Residues

The treatment with aqueous shoot extracts of ragweed (ST), root extracts (RT), and essential oil (EO) reduced root length and root mass, as well as the aboveground biomass (AGDM) of soybean, substantially, in the laboratory and in greenhouse experiments. Unlike soybean, the AGDM of wheat was not affected by any of the extracts, but we observed significant negative impact on the root growth of wheat seedlings in both trials. In contrast, the AGDM of maize decreased significantly, whereas root growth was clearly enhanced by ragweed shoot extracts and ragweed essential oil, but was not affected when exposed to ragweed root extracts. However, the strongest negative impact on all crops and ragweed itself was observed with ragweed residues, particularly when incorporated in soil. Furthermore, we found clear evidence that roots were more susceptible to growth inhibition than the AGDM and that there were differences in the sensitivity of the crops and ragweed.

In the present study, the most abundant compounds were sesquiterpenes, such as germacrene D and β-caryophyllene, sesquiterpene lactones, such as isabelin and psilostachyin, as well as monoterpenes, such as D-limonene, β-pinene, and myrcene. Refs. [26,28] already showed the growth inhibiting effect of sesquiterpene lactone isabelin on various crops, such as radish. Particularly, in the essential oil, we found high amounts of germacrene D. Studies on the effects of this sesquiterpene are available in the field of entomology, as this substance plays a vital role in the attraction, reproduction mechanisms, and oviposition of various insects of the order Lepidoptera, Coleoptera, and Diptera. Specific studies on the direct effects of this substance on other plants are—to the best of our knowledge—not available. Nevertheless, allelopathic effects of plant species having high concentrations of germacrene D have been demonstrated for goldenrod [35] or *Kundmannia sicula* [36].

In addition, it has been shown that α/β-pinene, D-limonene, and β-caryophyllene exhibit phytotoxic activity on various plant species. For D-limonene, negative effects, such as growth inhibition of shoots and roots, were reported for wheat, barley, and perennial ryegrass [37], as well as carrot and cabbage [38,39]. In contrast to the results of our study, Refs. [40,41] reported an inhibition of root growth in maize by inducing the production of malondialdehyde, which is an indicator of oxidative stress. For wheat and a legume species (*Pisum sativum*), it was shown that α/β-pinene have the potential to inhibit cell division in growing root tips [42]. In addition, it was demonstrated that α/β-pinene inhibited mitosis through interference with DNA synthesis in meristematic cells of *Brassica campestris* seedlings [43] and induced a reduction in chlorophyll content in rice (*Oryza sativa*), which suggests its negative impact on photosynthesis [44]. Of particular interest could be the high concentrations of β-caryophyllene, as [45,46] showed that this sesquiterpene can modulate jasmonic acid (JA)-mediated signalling, which is known as a stress-related hormone in plants. In this context, Refs. [46,47,48] demonstrated significant impact on root architecture and severe negative effects on root growth of *Arabidopsis thaliana* after the exposure to β-caryophyllene.

However, whether the observed impacts of ragweed were due to sesquiterpenes, monoterpenes, or combined synergistic effects of all the compounds could not be confirmed in the present study. Nevertheless, we found clear differences in the response behavior and the sensitivity of the crops and ragweed itself to ragweed extracts and residues, which underlines the findings of numerous studies performed with different ragweed extracts [49,50,51]. Especially, Ref. [52] showed that crops can even be negatively affected by the presence of chemical compounds of ragweed, without direct aboveground or belowground interaction with the plant or its extracts/residues. In a study on the effects of airborne volatile organic compounds (VOCs) of ragweed on soybean, wheat, and maize, it was shown that the exposure to airborne VOCs of ragweed was enough to reduce it, i.e., the AGDM of soybean, by 41%. Similar patterns were revealed with maize and wheat.

Particularly, ragweed reacted differently depending on the extract to which it was exposed. Root extracts and essential oil of conspecifics reduced AGDM. Root growth was only affected by root extracts in the laboratory experiment, but this effect was not observable when the root extract was incorporated in soil. In contrast, the exposure to shoot extracts had a clearly promoting effect on ragweed AGDM and roots which could be related to a sort of intraspecific competition effect. Ref. [53] investigated the sensitivity of ragweed to intraspecific competition. They showed that ragweed produced its highest individual plant biomass in high density monoculture. Furthermore, Ref. [52] found significant adjustment in the leaf mass fraction of ragweed when plants were exposed to airborne VOCs of conspecifics, assuming that VOCs emitted from a kin act as an early warning mechanism to avoid future mutual shading.

### 3.2. Effects on the Nodulation with Bradyrhizobium japonicum on Soybean Roots

Contamination with ragweed roots, as well as the presence of root extracts, impaired the nodulation performance of rhizobia, which can have severe effects on biomass production and yield of different legume species [15]. Studies dealing with the interaction between weeds and soybean, as well as how these weeds affect nodulation, are scarce. However, Ref. [24] showed that chemical components of ragweed had clear antibacterial activity, even in very dilute solutions against a broad range of bacterial strains. Especially, the sesquiterpene germacrene D and the sesquiterpene lactone isabelin were able to inhibit soil-borne bacteria and even human pathogenes [24,28,35,54]. In addition, it was shown that allelochemicals, such as α/β-pinene, can induce alterations in the production of reactive oxygen species (ROS), such as hydrogen peroxide (H_2_O_2_), which plays a major role in the infection process and the subsequent bacterial differentiation into the symbiotic form [23,55,56]. In this context, also, β-caryophyllene may have an impact on the nodulation process. As demonstrated by [57], β-caryophyllene shows rapid chemical reaction with different oxygen species, possibly generating new products that are themselves active microbial agents. Another role of β-caryophyllene involves protection against ROS. Ref. [58] showed that the sesquiterpene reacts very quickly with H_2_O_2_, leading to a severe reduction in the ROS-mediated signaling, which—in the case of our study—could be an explanation for the reduction in rhizobial nodules when ragweed compounds were present.

Additionally, on cellular level, it was shown that the presence of ragweed and its compounds, such as α/β-pinene, induced abnormal cell division and different kinds of nuclear alterations in the root tip meristematic cells, which subsequently negatively affected, i.e., the activities of the root system and the antioxidant enzyme [22]. Thus, the allelochemicals of ragweed can alter nodulation processes by interrupting the signaling pathway between soybean roots and rhizobial bacteria [59].

## 4. Materials and Methods

### 4.1. Plant Material

Aboveground biomass and roots of four different specimens of common ragweed were collected in July 2019 on four different sites in Austria and mixed together to avoid possible habitat/site effects in the plant material. Plant material was air-dried for 14 days at 40 °C and then kept in cool storage (4 °C) until it was further processed. Seeds of wheat (*Triticum aestivum*, cv. Capo), soybean (*Glycine max*, cv. Angelika), and maize (*Zea mays*, cv. ES Katamaran) were obtained from Saatzucht Probstdorf GesmbH, Austria. Mature, dry seeds of ragweed were collected in autumn 2019, on seven different sites in Austria and Germany, and mixed together to avoid possible parental or habitat/site effects. The seeds were air purified and placed at 4 °C in a dark refrigeration chamber until the beginning of the experiment.

### 4.2. Extract Preparation and Chemical Analysis

Essential oil was extracted from dried aerial parts (500 g) of common ragweed by hydro-destillation at atmospheric pressure, for 1 h, using a Clevenger-type apparatus (oil content: 0.1% per 0.5 mL of destillation). For the preparation of the aqueous shoot extract (stem + leaf), we used ultrasound extraction. Therefore, 100 g dry mass of ragweed shoots (stem + leaf) were mixed with 900 mL H_2_O for 30 min. For the aqueous root extracts, 1.800 g of fresh roots were steeped in 12 litres of 60 °C H_2_O for 15 min.

Chemical analysis of the oil was performed in our own lab by GC-FID (gas chromatography with flame ionization detector) using an Agilent 7890A gas chromatograph (Santa Clara, CA, USA), following an adapted protocol of [24]. Aqueous shoot and root extracts were analyzed by high-performance liquid chromatography (HPLC) at the Department of Molecular Science, Swedish University of Agricultural Sciences, Uppsala.

### 4.3. Laboratory Experiment

The potential inhibition of ragweed extracts and leaf tissue was studied by assessing its impacts on the crops seed germination and radicle elongation. The first experiment was conducted using Petri dishes (diameter: 90 mm) for the small seed species ragweed and wheat, as well as square Petri dishes (L × W × H: 120 mm × 120 mm × 17 mm) for maize and soybean. The initial aqueous extracts of ragweed shoots and roots served as stock solution, which was used in pure version in the experiments, but which was also diluted 10 and 100 times to prepare a set of decreasing extract concentrations (100–10–1%). The same was performed with the essential oil to obtain a 0.5% and 1% solution. In addition, dried ragweed shoots were cut into small pieces with a grinder and weighed into 0.32, 0.64, and 0.95 g portions (equals to 0.5, 1.0 and 1.5 t per ha^−1^). The amount used is similar to that used in other allelopathy experiments on Asteraceae species [60,61].

An amount of 100 seeds of each crop species and ragweed were placed on two-layer filter paper in the Petri dishes in four replications for each treatment. An amount of 2 mL of the different extract solutions was added to the Petri dishes. To the square Petri dishes, we added 12 mL of water. Prior to adding the residues, 2 and 12 mL, respectively, of tap water, was filled in the Petri dishes to secure humid conditions, which are necessary for germination. In total, 100 seeds were tested with each extract × concentration combination. Petri dishes with tap water only were used as control. Petri dishes were placed in an incubator with 12 h of full light at 23 °C and 12 h darkness at 15 °C for 24 days. During the bioassay, filter paper was kept wet by periodical adjustment of extract loss. The number of germinated seeds was recorded three times a week. The radicle length of the crop and ragweed seedlings was measured 10 and 17 days after beginning of the bioassay (adapted protocols from [26,62].

### 4.4. Greenhouse Experiment 1

In the first greenhouse trial, the same extract × concentration × plant species combination was tested in pots (5 cm × 5 cm × 5 cm) filled with 200 g of potting soil consisting of sand, peat (1 part quartz sand, 3 parts ecological, peet-free universal soil “Ökohum”; pH 6.9), n 30 replications (in total: 1440 pots). To each pot, we added 75 mL of the respective treatment. The holes on the bottom side of the pots were covered with a clay ball to secure a natural run-off and gas exchange while avoiding extensive leaching. For the residue treatment, the soil in the pots was weighed, and 1, 5, and 10% grinded ragweed dry matter was mixed into the soil. In each pot, three seeds of wheat, maize, soybean, or ragweed were planted to avoid bias in the results due to nonviable seeds. If all seeds germinated, two seedlings were removed immediately to avoid competition effects in the pots. All pots had saucers to prevent contamination of leachate from other treatments and were rotated every fifth day to account for spatial variation. Temperature and light regime in greenhouse were adapted to the regime in the laboratory (12 h full light at 23°/12 darkness at 15 °C). Throughout the experiment, pots were kept wet with tap water by using a hand sprayer. Germination rate, shoot length, root length, and aboveground and belowground dry matter were measured on 10 plants at 10, 17, and 24 days after seeding. Roots and aboveground biomass of all plants were separated at the growing point, dried at 105 °C for 24 h, and weighed.

### 4.5. Greenhouse Experiment 2

To test the effect of different soil types in combination with the extracts on the infection potential of soybean with nitrogen-fixing *Bradyrhizobium japonicum*, a second greenhouse trial was implemented in pots (8 cm × 8 cm × 8 cm), which were filled with 600 g of potting soil (pH 6.9), sand (pH 7.1), or arable soil from a surrounding field, which was classified as a chernozem of alluvial origin (pH: 7.6) in 10 replications. Due to the severe effect of root exudates of ragweed on soybean root growth, we implemented two different treatments (10 replicates per treatment and soil type), consisting of undiluted ragweed root extracts and pots filled with root residues of ragweed. Therefore, we planted one ragweed plant in each pot and let them grow until the emergence of male inflorescence. In this developmental stage, the secondary metabolite content in the plant is its highest [26]. Afterwards, aboveground parts of ragweed were removed. Soybean was planted in the pots where the ragweed roots remained. In addition, we included untreated control pots in the experiment. Before seeding, soybean seeds were inoculated with *Bradyrhizobium japonicum* (RWA Raiffeisen Ware Austria, Korneuburg, Austria), a nitrogen-fixing bacterial species that forms root nodules specifically on soybean roots but is not sufficiently abundant in Central European soils. After 50 days, all plants were removed from the pots, roots were washed out manually, and nodules were counted.

### 4.6. Statistical Analysis

Data analysis was performed using software R, Version 4.0.5 (R Core Development Team, 2021). For the graphical visualization of the results, Sigma Plot, Version 14.5 (Systat Software, 2022) was used. Analysis of variance with subsequent multiple comparisons of means according to Tukey were performed (significance level α = 0.05). Shapiro Wilk’s test was used to test the normal distribution of data, and Levene’s test was used to check homogeneity of variances. If normal distribution was not given, a Kruskal–Wallis ANOVA on ranks was performed. If homogeneity of variances was not given, a statistical analysis was executed using Welsh’s unequal variances *t*-test. When preliminary analysis showed that data transformations were not needed to meet basic assumptions for regression analysis, dose-response relationships were calculated using R-package “drc” [53]. Model selection was based on the Akaike Information Criterion (AIC) value. To identify the most parsimonious model based on the lowest AIC value, the AIC difference between the different candidate models was computed. As a rough rule, Ref. [63] proposed that, for models wherein Δi ≥ 10, the same receive substantial support. Fitted equations were used to calculate the doses required to reduce radicle growth by 20% (ED20) and 50% (ED50).

## 5. Conclusions

The results of our study imply that chemical compounds of ragweed can alter not only the aboveground and belowground growth performance of crops, but they are also able to alter the performance of soil microorganisms. Thus, the chemical interaction among plants is an important functional component not only in natural plant communities, but also in agricultural systems. It may, therefore, play an important role in future crop research. In the present study, effects of ragweed extracts and residues were related to the concentration, but also to the medium (Petri dish vs. soil) used. This implies that soil can have retarding and promoting effects of chemical compounds, which should be taken into account in further research. In addition, we found clear differences in the response between crops (promoting and inhibiting effects). As our experiments were conducted on seedlings under greenhouse conditions, we encourage future field trials to obtain more information on the effective period in which the extracts and residues have an influence on the development of the crops. With a view to an improved crop production, this study can contribute to the adjustment of weed management regimes to avoid soil contaminations, particularly with ragweed residues, which can have substantial impacts on the seedling development of crops.

## Figures and Tables

**Figure 1 plants-12-01768-f001:**
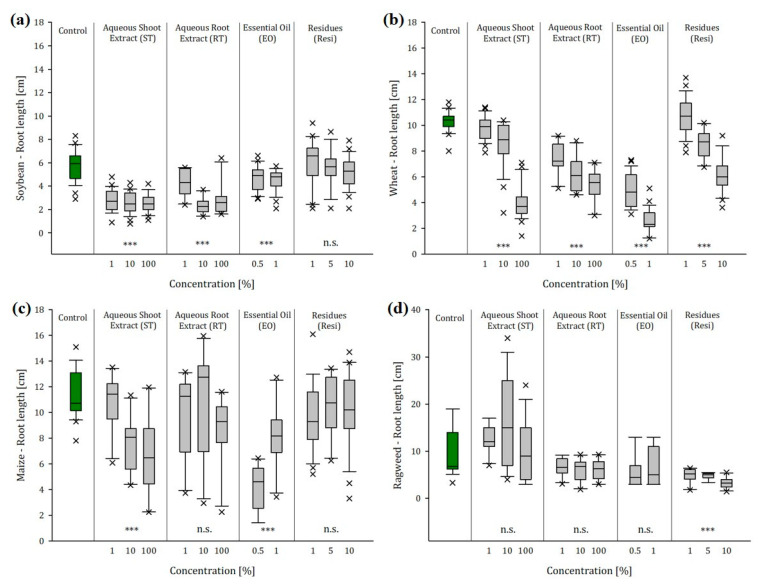
Root length (cm) of (**a**) soybean, (**b**) wheat, (**c**) maize, and (**d**) ragweed in dependence of the concentration of aqueous leaf extract (ST), aqueous root extract (RT), essential oil (EO), and residues (Resi) of ragweed 24 days after seeding (*n* = 10, significance levels: n.s. = not signficant, *** *p* < 0.001 refer to the untreated control).

**Figure 2 plants-12-01768-f002:**
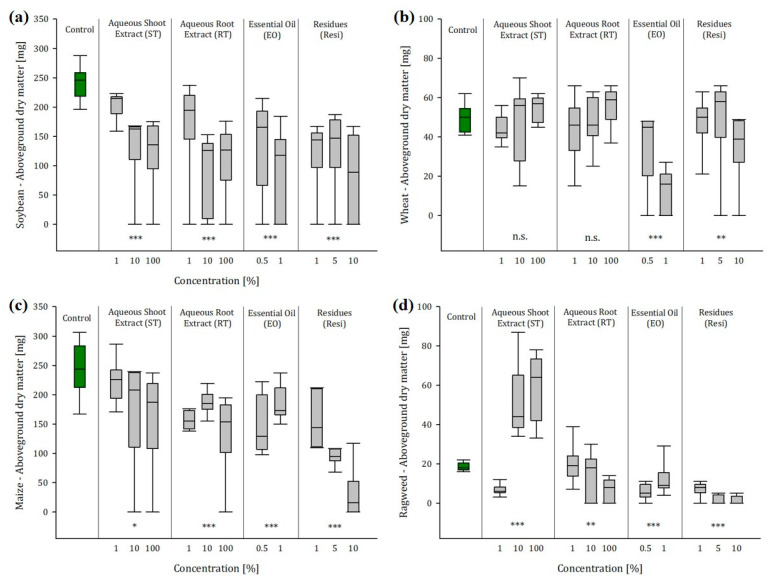
(**a**–**d**) Aboveground dry matter (mg) of (**a**) soybean, (**b**) wheat, (**c**) maize, and (**d**) ragweed in dependence of the concentration of aqueous leaf extract (ST), aqueous root extract (RT), essential oil (EO), and residues (Resi) of ragweed 24 days after seeding (*n* = 10, significance levels: n.s. = not significant, * *p* < 0.05, ** *p* < 0.01, *** *p* < 0.001 refer to the untreated control).

**Figure 3 plants-12-01768-f003:**
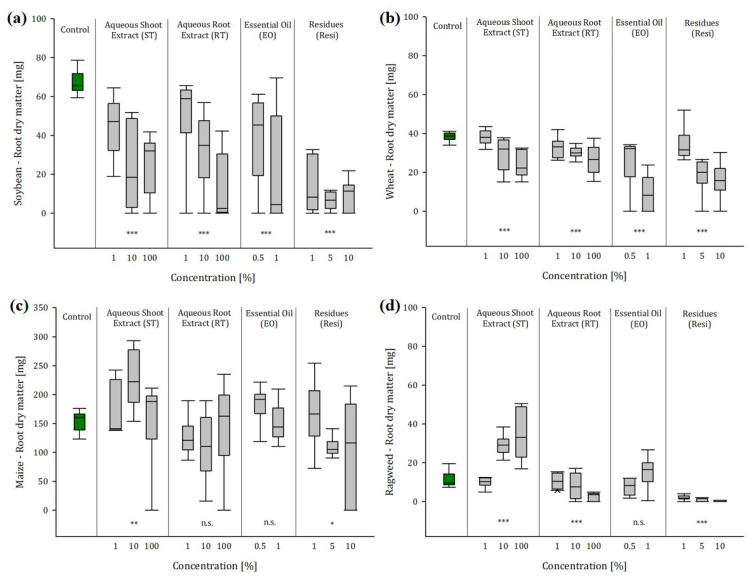
(**a**–**d**) Root dry matter (mg) of (**a**) soybean, (**b**) wheat, (**c**) maize, and (**d**) ragweed in dependence of the concentration of aqueous leaf extract (ST), aqueous root extract (RT), essential oil (EO), and residues (Resi) of ragweed 24 days after seeding (*n* = 10, significance levels: n.s. = not significant, * *p* < 0.05, ** *p* < 0.01, *** *p* < 0.001 refer to the untreated control).

**Figure 4 plants-12-01768-f004:**
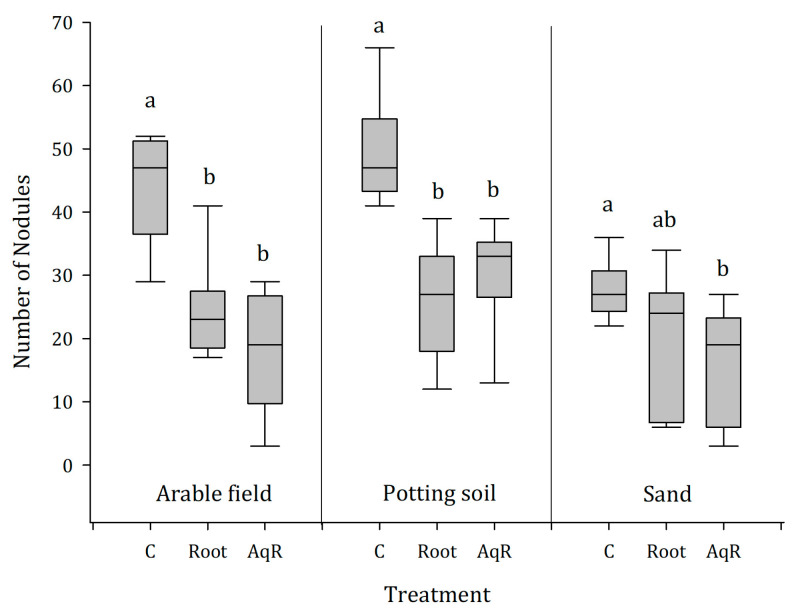
Number of nodules in relation to the presence of ragweed roots (Root), presence of undiluted root extract of ragweed (AqR), or untreated (control = C) in three different substrates (soil from arable field, potting soil, and sand); *n* = 10, different letters indicate significant differences.

## Data Availability

Due to privacy restrictions, we cannot publish the raw data. If the data are needed for further research, we are willing to send the data upon request. Therefore, please send a short statement why and for what purpose the data is needed to the corresponding author.

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
