# Peer review of "Extracts and Residues of Common Ragweed (Ambrosia artemisiifolia L.) Cause Alterations in Root and Shoot Growth of Crops"

_plants, 2023, doi:10.3390/plants12091768_

Round 1
Reviewer 1 Report
This study seems to me to be well structured and the thesis respected and adequately discussed. I therefore have no objection to its publication.
Author Response
Thank you very much for your very positive feedback!
Reviewer 2 Report
I’ve read with attention the paper. The background and aim of the study have been clearly defined. The methodology applied is overall correct, the results are reliable and adequately discussed. However The novelty of this research work is very low, the compounds under study have well known allelopatic activity, only the crops considered are different from published works.
- The authors should add some biochemical aspects. With the addition of biochemical attributes paper can be considered as review. But with out this paper is not recommended for publication.
Author Response
Thank you very much for your valuable comment. We changed the discussion significantly with more emphasis on the mode of actions of the chemical compounds which could have caused the alterations in shoot and root growth.
Reviewer 3 Report
When taking extracts, if chemicals are to be used, they need to be treated for at least 48 hours, and for water extract it should have been even longer. You took a water extract and treated it for a short time. This type of extraction method was not appropriate. Again, the amount of water added to the petri dishes and pots is quite low. I wonder how germination was achieved with this low amount of water. You should make sure that the references used in the study are new. It would be appropriate to add some more recent literature.
Author Response
Thank you very much for your effort and your very valuable comments. Please find the justifications in the attachment.

Round 2
Reviewer 2 Report
Dear Author, With the modification of discussion, the paper has a clear story now. And hence can be accepted for publication.